# Tuberculosis Disability Adjusted Life Years, Colombia 2010–2018

**DOI:** 10.3390/tropicalmed7090250

**Published:** 2022-09-18

**Authors:** Laura Plata-Casas, Oscar Gutierrez-Lesmes, Favio Cala-Vitery

**Affiliations:** 1Faculty of Natural Sciences and Engineering, University Jorge Tadeo Lozano, Bogota 111711, Colombia; 2School of Public Health, Faculty of Health Sciences, University of the Llanos, Villavicencio 500003, Colombia

**Keywords:** tuberculosis, disability-adjusted life years, Colombia

## Abstract

Estimating the burden of tuberculosis disease is relevant for assessing and identifying population health status and progress in policies and programs aimed at epidemic control. The objective of this study was to estimate disability-adjusted life years attributable to Tuberculosis in Colombia 2010–2018. A longitudinal descriptive study was conducted. The variables, sex, age groups and origin were studied. This study included 110,475 cases of morbidity and 8514 cases of mortality. Indicators of years of life lost, years of life with disability and disability-adjusted life years at the subnational level were determined using the methodology of the World Health Organization. With the results of this last indicator, an epidemiological risk stratification was carried out. The DALY rate of the study period was 684 (95% CI 581.2–809.1) per 100,000 inhabitants. According to sex, 68.4% occurred in men; for every DALY in women, 2.21 occur in men. People of productive age (15 to 69 years) account for 56% of DALYs. Amazonas (1857.1 CI 95% 1177.1–2389.6) was the territorial entity with the highest rate. A total of 51.5% of the territorial entities of departmental order of the country are of high burden for Tuberculosis. For the first time in Colombia, a comprehensive assessment of the status of the disease burden at a subnational (departmental) territorial level attributable to Tuberculosis is being carried out using the updated World Health Organization methodology. The results obtained allow us to specify that there is a knowledge gap in terms of the realization and clear understanding of the burden of the disease in Colombia. There are territorial gaps that are necessary to know in order to plan, develop, implement and redirect policies to improve health and eliminate disparities according to the territorial context.

## 1. Introduction

Tuberculosis (TB) is a preventable and curable infectious disease caused by Mycobacterium tuberculosis [1]. It produces significant morbidity and is one of the top ten causes of mortality in the world [2]. It is estimated that about one-fourth of the world’s population is latently infected [3], who, as reservoirs, can develop active disease at any time in their lives. Symptoms may include cough, fever, night sweats and weight loss. These symptoms can be mild and last for several months, during which time a patient with TB can infect between 10 to 15 people by close contact [4]. For TB in Colombia, an epidemiological surveillance system [5] is in place (SIVIGILA), the National Tuberculosis Prevention and Control Program (PNPCT) takes a comprehensive approach, and there is a broad regulatory framework [6,7]. As an event of interest in public health, it has protocols for tuberculosis of all forms and sensitive to drugs and for drug-resistant tuberculosis [8,9]. These protocols include aspects related to TB/HIV confection or related to immunosuppressive pathologies or treatments. Its surveillance includes operational case definitions (clinical, laboratory or epidemiological link confirmation), classification based on history of previous treatment of tuberculosis (new or previously treated case), classification according to anatomical location of the disease (pulmonary, extrapulmonary) and classification based on HIV test status (person with tuberculosis and HIV, person with tuberculosis and without HIV, person with tuberculosis and HIV status unknown). Regarding drug-resistant tuberculosis, in addition to the configuration of the case, it includes the classification according to type of drugs received (treated with first-line drugs, has not received first-line drugs), the classification of the case according to admission condition (after relapse, patient with treatment after loss to follow-up, after failure) and the classification of the case based on the type of drug resistance (monoresistance, multidrug resistance MDR, polyresistance, previous extensively resistant pre-XDR, resistant XDR, resistance to Rifampicin).

Risk factors for TB include ambient temperature and relative humidity [10], COVID-19 [11], age, poverty, diabetes, malnutrition and comorbidities such as HIV [12]. Those suffering from tuberculosis face a continuous disability [13] with subsequent sequelae of high impact.

In 2019, TB affected around 10 million patients and caused 1.2 million deaths worldwide; these figures undermine the World Health Organization (WHO) global goals to reduce the incidence and mortality of this disease, with progress in its care and prevention being very slow [14]. According to the 2019 Global Burden of Disease Study [15], TB at all ages is among the causes that had the greatest decrease in the absolute value of the number of healthy life years lost DALYs between 1990 and 2019. According to this study, in 2019, 2540 million DALYs were lost, of which 1.9% was due to TB, with the DALY rate for TB being 590.4 (95% CI 536.8–646.4) per 100,000 inhabitants [15].

According to the global TB 2020 report [2], the Americas contributed 2.9% of cases in 2019, and the incidence is slowly increasing due to the upward trend that is occurring in Brazil, a country bordering Colombia. Estimated deaths for the region were 27,000, and lethality was 7% compared to 14% globally [16]. In Colombia in 2020, 12,582 cases were reported, which corresponds to 6.6% of the cases in Latin America, with a comparative decrease of 19.2% compared to the previous year, being considered a high-burden country; its treatment coverage is 79.4%, and 5% of the reported cases correspond to the indigenous population [16]. For Colombia in 2019, the age-standardized DALY rate for both sexes was 64.58 per 100,000 inhabitants [10].

Global initiatives have been developed to combat TB. In 2014, WHO member countries made a commitment to end the epidemic through the adoption of the End TB Strategy. With the United Nations Sustainable Development Goals (SDGs), the Moscow Declaration to End this Disease was adopted by the Member States of the World Health Assembly in May 2018 [17]. These science-based initiatives promote interventions to address the obvious risks of contracting and dying from TB, aiming to end the epidemic by 2030, significantly reducing the incidence, deaths and costs faced by patients and their families, and implementing measures that promote equality and protect rights.

The indicator of disability-adjusted life years or healthy life years lost due to DALY diseases or injuries, proposed by Murray and López, authors of the Global Burden of Disease study [18], allows the comparison of diseases with each other and populations, synthesizing epidemiological mortality data in a single value, morbidity and disability. The advantage of using it for health planning is that it allows us, over time, to observe the evolution of the health of a population or the magnitude of a health problem, and it supports the definition of priorities in terms of interventions and the evaluation of the impact of these and uses these results to define priorities and guide the allocation of resources.

DALYs constitute the burden of disease and consist of the sum of years of life lost due to premature death (YYL) and years of life with disability (YLD). YYLs are the years that a person stops living when they die before reaching a theoretical life expectancy [19]. This theoretical expectation is met with assumptions such as having an optimal state of health, not being exposed to risk factors, not presenting injuries and having access to adequate and timely health services. YLDs measure disability as a deviation from health in any of the domains (mobility, self-care, activities of daily living, pain, discomfort, anxiety, depression, social participation, cognition, among others). These domains are classified as listed in the International Classification of The Functioning of Disability and Health ICF and are consistent with the codes of the International Classification of Diseases ICD10 and use disability weights [20]. The WHO updated the metric of this indicator [21] and through it periodically advances the study of disease burden. For TB, the WHO constructs the indicator for each country but does not disaggregate it at the subnational level for Colombia.

The use of the DALY indicator in Colombia has been scarce [22] and has been used with the methodology of the 90s [23,24], which, in general, does not allow comparability between studies. The methodology that constitutes the burden of disease is not institutionalized for comprehensive health planning in the country [25]. Determining and understanding the burden of TB disease is critical to assessing and identifying health status and progress in ending the epidemic, as well as in reporting progress in the implementation of related public policies and disease control programs, and in directing the health system.

Colombia is a country located in South America with internal division into 32 territorial entities of departmental order and a capital district. For health, it has a broad set of rules that establish affiliation to the General System of Social Security in Health [26] and the execution of activities of prevention, surveillance and control of TB in a decentralized manner [5,27,28,29]. It has a national TB control and prevention program that provides technical and operational guidelines [6] for programs at the departmental, municipal and institutional levels to operate them.

The Ten-Year Public Health Plan 2022–2031 [7] is a state policy that guides public health for the next decade. This plan considers TB as a priority and establishes the reduction of 50% of the mortality rate caused by TB (baseline of 1.93 × 100,000 inhabitants) and the achievement of 90% of successful treatment (baseline 71%). It does not consider the DALY as an indicator for the event.

Based on the application of the updated methodology for the determination of DALYs according to ICDX coding, using the national population projections for the years 2005–2020 of the National Department of Statistics (DANE in Spanish), the objective was to estimate the disability-adjusted life years (DALYs) attributable to TB in Colombia 2010–2018 and the epidemiological stratification of subnational risk according to this indicator. The results show that in Colombia the control of TB remains a challenge and reveals wide disparities in the burden of the disease at the subnational level.

## 2. Materials and Methods

A descriptive epidemiological study was conducted for Colombia based on the historical cohort 2010–2018. Data on disease cases and mortality from TB were obtained from the Integrated Information System for Social Protection (SISPRO in Spanish). This system consolidates the non-fetal morbidity and mortality data from the health service provider network, which are obtained from the Individual Registry of Service Provision (RIPS in Spanish) and the Vital Statistics System (RUAF in Spanish).

The data analyzed were obtained, anonymized and validated by the Ministry of Health and Social Protection upon consultation requested from that entity for all forms of TB, by sex, age and department of origin, according to ICDX codes (A150-A171, A178-A199, J65X, K230, K673, K930, M011, M490, M900, N330, N740, N741, O980 and P370), thus controlling the garbage codes.

In the Colombian context, the RIPS, which consolidates morbidity, is a set of data regulated since 2007 [30], which contains information related to the provision or provision of health services and technologies to users. They are generated, validated and sent by health service providers and have a review process in content and consistency [30]. Mortality data from death certificates issued by service providers and the National Institute of Legal Medicine and Forensic Sciences were requested for the codes according to the basic cause of death. The basic cause of death contrasts the causal chain reported on the medical death certificate with the cause coded by DANE.

We used the inclusion criterion of all cases reported in the country during 2010–2018 at any age with morbidity and basic cause of death from any type of TB. The exclusion criteria were duplicate records and records in which the place of residence did not correspond to a department of Colombia and fetal death records. Based on these criteria, 1,100,632 records were selected for morbidity and 8514 cases for mortality, which met the characteristics required for the analysis. A quality control process was carried out to detect typos and data loss, and two new databases were built, one of morbidity and one of mortality. No sampling was required because all the records in the database were taken into account.

### 2.1. Variables Studied

The YLL, YLD and DALY indicators by sex, origin and for 17 five-year age groups (0–4, 5–9, 10–14, 15–19, 20–24, 25–29, 30–34, 35–39, 40–44, 45–49, 50–54, 55–59, 60–64, 65–69, 70–74, 75–79 and 80 and more) were constructed for the first time for Colombia with the updated WHO methodology.

### 2.2. Statistical Analysis

We proceeded to the identification of the sociodemographic characteristics with respect to the morbidity and mortality data. The variables were described using descriptive statistics. The SPSS™ program (Bogotá, Colombia), version 23 owned by the University of Los Llanos, was used. The construction of indicators by territorial entities for each year and for the study period was carried out using the number of cases as a numerator and the population as denominator according to the DANE 2005 census report, by sex and age group. For the study period, the rates were calculated considering the mid-term population. All estimates were calculated as crude counts and rates per 100,000 people and with 95% confidence intervals (CI). The 95% confidence intervals for these indicators were determined by the bootstrap technique using the XLSTAT program in 1000 samples with bias corrections. The calculation of the synthetic indicators YLD, YLL and DALY was carried out by age group, sex and territorial entity, using the variable transformation tool in the SPSS™ program (Bogotá, Colombia), version 23, owned by the University of Los Llanos. This is considering the number of deaths (YLL) and the number of cases of disease (YLD), multiplying them by their respective factors (weight by disability and weighting factor by age). The codes were as follows: DATASET COMPUTE YLD = CASES × DISABLEDWEIGTH. EXECUT and COMPUTE YLL = CASES × WEIGHTINGFACTOR. EXECUTE. Finally, the calculation of the DALY was performed using the SPSS™ program transforming/calculating variables with the YLD and YLL previously estimated, with the code: COMPUTE DALY = YLD + YLL. EXECUTE.

#### 2.2.1. Years of Life Lost (YYL): Deadly Effects of TB

For the calculation of the YYL [31] in the DALYs metric, a standard life expectancy should be used to be comparable between different populations, which is 92 years. The expression for calculating YYL is Equation (1):(1)$$\mathrm{YLL}=\overset{z}{\underset{a}{\sum }}D_{c,a,s,t}e_{x}^{*}$$
where YLL is total years of life lost, D is the number of deaths due to the cause (c) in the age group (a), in sex (s), and year t. $e_{x}^{*}$ is the life expectancy at each age (the weighting factor is derived from the standard life expectancy (SLE) recommended by WHO, based on a 92-year-old SLE). The number of tuberculosis mortality cases was estimated for each territorial unit according to sex and five-year age groups.

#### 2.2.2. Years of Life with Disability (YLD): Non-Fatal Effects of TB

For Years Lived with Disability (YLD), the calculation expression is Equation (2):(2)$$\mathrm{YLD}=\overset{z}{\underset{a}{\sum }}{\mathrm{dw}\;P}_{c,a,s,t}$$

In the above formula, YLD is the total years lived with disability, (a) and (z) refer to TB classifications, (dw) is the disability weight, P is the prevalence of the disease or injury (c), in the age group (a), according to sex (s), and year (t), according to GBD 2019 disability weights for each health state. Regarding the disability weights for the different health states, the updated weights of the Institute for Health Metrics and Evaluation IHME of the University of Washington of the GBD 2019 were used, an update recognized by WHO to be applied in the measurement of the burden of the disease. TB prevalence was estimated for each territorial unit by sex and five-year age groups.

#### 2.2.3. Disability-Adjusted Life Years (DALYs) or Healthy Life Years Lost to TB

The DALY indicator corresponds to the sum of years of life lost due to premature death (YYL) and years of life with disability (YLD) by sex and age [32]. The calculation expression is Equation (3):DALY_c,a,s,t_ = YLD_c,a,s,t_ + YLL_c,a,s,t_(3)

In the above expression, DALY(c,a,s,t) is the total disability-adjusted life years, YLD is years lived with disability, YLL is years of life lost, for the cause (c) in the age group (a), in sex (s), and year (t).

Once the DALY indicator was established, the mean was calculated to define a comparison value; from this value, the 75th percentile of these data was calculated, and two categories were defined: high load is equal to or greater than the 75th percentile, and low load is less than the 75th percentile, a methodology used in the Colombia Tuberculosis-Free Strategic Plan [33]. Based on this classification, the departmental territorial entities were categorized for the study period.

### 2.3. Bias Control

The theoretical assumption of attributing mortality to the basic cause was used to control for selection bias. The under-registration information bias could not be corrected due to the lack of measures of integrity of the vital statistics system [34]. With the inclusion of all reported cases from all departments of the country along with the reports of the Colombian Institute of Legal Medicine and Forensic Sciences, the health care bias was mitigated.

### 2.4. Ethical Considerations

This study complied with all the requirements of Resolution 8430 of 1993 [35] for health research in Colombia. The method of collection was documentary, confidentiality was safeguarded by not using names or identity numbers.

## 3. Results

In the study period, 110,475 cases of morbidity and 8514 cases of mortality were reported (Table 1). Regarding the distribution by sex, men were the most affected (62.6% in morbidity and 70% in mortality). In terms of age groups, people of productive age (15 to 69 years) presented 81.3% morbidity and 59.8% mortality. The territorial entities with the highest contribution of cases were Antioquia (19.9% for morbidity and 15.4 for mortality), Valle (14.6 for morbidity and 13.9 for mortality), Bogotá (7.7% for morbidity and 10.4 for mortality) and Atlántico (7.1% for morbidity and 8.6% for mortality). The departments of Guainía and Vaupés have reports of low morbidity in several years of the study period.

### 3.1. Years of Life Lost (YYL): Deadly Effects of TB

The rate of YYL in Colombia during the study period was 606.8 (95% CI 491.6–713.8) per 100,000 inhabitants. The years with the highest rate were 2010 and 2018 with 75.8 (95% CI 60.5–88.5) and 72.7 (95% CI 58.6–88.6), respectively. Regarding sex, 69.1% of the total YYL in the country were in men; for each YYL in women, there were 2.29 in men (Table A1). When the loss was compared according to the age group to which the deceased belonged, it was highlighted that people of productive age (15 to 69 years) are the most affected, comprising 76.2% (Figure 1).

When premature mortality was discriminated against at the subnational level, the territorial entities that stood out for presenting the highest rate during the study period were Amazonas (1561.1, IC 95% 996.7–2232.5), Risaralda (1167.6 IC 95% 891.4–1483.9), Atlántico (1081.5 IC 95% 891.7–1299.7), Guainía (1069.1 IC 95% 524–1789.2) and Meta (1048.4 IC 95% 858.7–1257). Vichada (191.6 CI 95%13.4-468.7) and Guaviare were the entities with the lowest rate (119.7 CI 95% 0–376.3). Guaviare, Vaupés and Vichada in several years of the study period report zero deaths from TB (Table A1) in women.

### 3.2. Years of Life with Disability (YLD): Non-Fatal Effects of TB

The rate of the study period was 77.2 (95% CI 64.8–90.6) per 100,000 inhabitants. The years with the highest rate were 2016 (9.2 95% CI 7.8–10.5) and 2017 (9.2 95% CI 7.7–10.9). Regarding sex, men lived longer in a suboptimal state of health, with 62.6% of Colombia’s total ADLs; for each YLD in women, 1.71 are present in men (Table A1). When the non-fatal effect of TB was compared according to the age group, it was highlighted that people of productive age (15 to 69 years) were the most affected and concentrate 81.2% of these (Figure 2). At the subnational level, the territorial entities with the highest rates during the study period were Amazonas (296.1 CI 95% 255.1–348.3), Guainía (232.1 CI 95% 187.3–290.9), Risaralda (149.6 CI 95% 121.8–174.9), Chocó (148.5 CI 95% 123.7–178.8) and Meta (127.1 CI 95% 103.4–150.7). Boyacá (25.3 CI 95% 20.1–31.6) and Sucre (24.8 CI 95% 20.2–29.6) were the entities with the lowest rates (Table A1).

### 3.3. Disability-Adjusted Life Years (DALYs) or Healthy Life Years Lost to TB

The sum of the years of life lost due to premature mortality and the years lived in suboptimal health states, generated by TB constitute the burden of disease for this event in Colombia. The rate of the study period was 684 (95% CI 581.2–809.1) per 100,000 inhabitants. The years with the highest rate were 2010 (84.3 95% CI 72.1–100.5) and 2018 (81.4 95% CI 63–100). According to sex, 68.4% occurred in men; for each DALY in women, 2.21 were present in men (Table A1). The differences between the sexes became smaller in the older age groups. By age group, people of productive age (15 to 69 years) comprise 56% (Figure 3).

The health gap in the study period of each of the departments of Colombia, due to the loss of years of life, determined by the DALYs was greater, in magnitude, in the territorial entities of Amazonas (1857.1 CI 95% 1177.1–2389.6), Risaralda (1317.1 CI 95% 982.6–1683.8), Guainía (1301.2 CI 95% 581.6–2127.6), Atlántico (1189.6 CI 95% 988.1–1392.5) and Meta 1175.6 CI 95% 939.2–1363.4), while Boyacá (227.4 CI 95% 172.9–301.7) and Guaviare (151.8 CI 95% 58.5–308.5) had the lowest rates (Table A1). In the analysis of DALY rates by sex and age group, in each department, it was found that rates were higher in men, and the rates increased in both sexes as age increased. In total, 51.5% of the territorial entities of departmental order of the country were of high load for TB (Figure A1).

## 4. Discussion

In this article, we present for the first time in Colombia the most comprehensive assessment of the subnational status of the disease burden attributable to TB, measured for the first time in the country with the updated WHO methodology, highlighting the spatial distribution over a period of 9 years and contemplating its complexity as a territory. The burden of disease attributable to TB provides important arguments for reducing subnational gaps, achieving SDG targets and developing or modifying policies to reduce them.

What was found in the study against the greater involvement in men agrees with available evidence such as the global study of disease burden [36], Lee S et al. [37] and Rumisha et al. [38]. This evidence indicates that men have a higher morbidity and mortality than women in low- and middle-income countries such as Colombia due to reasons that may be related to roles, sociocultural behaviors, biological aspects [39], risk factors such as smoking and alcohol consumption and the lower probabilities of seeking or having access to TB care [40], contributing to worse treatment outcomes [41].

These differences may also be due to the fact that men have fears or anxieties that intersect with expectations of masculinity given the social pressure to ignore the symptoms to remain physically and financially strong and able to satisfy leadership roles [42]. In addition to the above, low use in health services, seeking health care with more advanced stages of the disease to avoid the diagnosis of a serious infection such as TB, low adherence to treatment [43], alcohol consumption [44], among others, may be other causes of sex differences.

Regarding age, our results are consistent with what was reported by Martial et al. [45], who found that, for those aged between 15 and 54 years, there is a high risk of incidence, and those who are 15 to 49 years old have a high risk of mortality, which is influenced by factors such as infection by the human immunodeficiency virus (HIV), smoking and excessive alcohol consumption (unhealthy lifestyle), among others. The increase in rates in both sexes as age increases may be due to the appearance of medical states such as diabetes [46] and other comorbidities that weaken the immune system, uncommon symptoms that make diagnosis difficult, and the increased risk of developing more severe and atypical forms of TB [47].

It is essential to consider that the age groups mainly affected in Colombia are those that represent the country’s workforce. Although this research did not contemplate an assessment of the cost of the TB epidemic on economic wellbeing, it is described that this disease adversely affects the labor force, depresses household savings and disrupts local economies [48].

In the context of the COVID-19 pandemic, knowledge about the concomitant infection of these two events is limited [49] and can generate complex results of disease burden, given the involvement of immunity [50,51], whose limited response can contribute to the deterioration, worse evolution and death of the affected person [52]. In addition to the above, the COVID-19 pandemic dramatically affected the provision of essential health services for TB [53]. This effect was due to factors such as the reassignment of health workers to other areas, the closures of the health system due to high demand due to COVID-19 [53], interruptions in care services for TB [54], delays in diagnosis, increased vulnerability due to the impact of the economy [55] and job loss, among others. Modeling studies have suggested that there may be an effect of increasing deaths from this cause [56] of up to 20% in the next 5 years [57], which would aggravate the impact on the population and the family and country economy. Morbidity may also increase due to increased opportunities for home transmission [57]. Although the temporality of this study did not include the timing of the COVID-19 pandemic, it is suggested to continue studying the phenomenon with respect to the impact of the pandemic on the burden of TB disease. In addition, it is important to evidence this impact not only seen from the factors of transmissibility by contact at home and service interruptions, but from the shared dysregulation of immune responses along with the severity of COVID-19 and the progression of TB disease [58]. Based on this impact assessment, it might be necessary to consider recalibrating the objectives set at national and international level.

As reflected in the data in our study, the rates of YYL, YLD and DALYs per year for Colombia are fluctuating. There are no previous national studies to compare the rates of YYL, YLD and DALYs per TB at the country level and subnationally. For Colombia, the global load study for TB reported a rate of YYL of 568.7, YLD of 82.4 and DALYs of 516.7 per 100,000 inhabitants during the study period [59]. In the case of YYL, our data are higher, and in the case of YLD and DALYs they are lower, which may be related to the sources of information.

Although social determinants of health are shared with bordering countries, Colombia has the lowest rate of DALYs for TB [31]. Brazil has different contexts of social vulnerability, and these populations show a heterogeneous temporal and spatial pattern [60], showing high-risk spatio-temporal conglomerates like what is present in Colombia. The above shows the need for integrated public policies acting both on social vulnerability and on the guarantee of a universal, free and quality health system. The rate of DALYs in Colombia is higher than that of Korea, which has the highest incidence of TB among the countries of the Organization for Economic Co-operation and Development (OECD) [37]. The possible explanatory routes related are latent tuberculosis infection, growing population with diabetes [61] and high elderly population, among others.

In Colombia, DALY rates have remained stable, in line with the latest reported global burden of disease study [31]. The data show the absence of a visible trend in disability, which could be due to the availability of data on the severity of the event, and they highlight the need to quantify the effect of health service interventions that modulate it. These rates in the country are low and should lead to an adequate assessment of the burden of disease during and after the disease. TB can cause temporary or permanent disability, arising from the disease process itself or from side effects related to its treatment.

This should include greater programmatic attention that can prevent TB, encourage the active search for cases to accelerate diagnosis and interventions that prevent, mitigate or repair lung damage, the effect of medications on mental health, stigma or self-stigma [62] and hearing and neurological disability [63], among other factors. Colombia must begin to establish the impact of these aftermaths. Describing the spectrum of related disabilities is essential for both service delivery and policy formulation, as disability from this event can become a large component of the burden of disease and health expenditure. It is suggested to reconsider the definition of treatment success, making it broader and not relating it only to the negativization of diagnostic tests and the termination of treatment.

The concentration of cases in some territorial entities surrounded by others with low or no occurrence may indicate under-registration in the entities of origin [64] and centralization in the receiving entities that act as reference and counter-reference centers for the provision of services in the country. There are entities such as Guaviare, Vaupés and Vichada with periods of low or no YYL reports, especially in women and low rates of YLD. These entities have high rurality, low access to molecular diagnostic technologies, difficulties of geographical accessibility and varied populations, especially indigenous ones. Gele et al. highlighted limited access to TB services as one of the main barriers to seeking medical care [65]. Other studies also show that factors such as living in rural areas [61], low income and unemployment, all of which are present in the areas mentioned, can affect timely diagnosis [66]. With regard to the indigenous population, it was important to take into account the effects of colonization and policies rooted in racism [67], and not only race, as decisive factors [68].

This finding is considered significant given the aforementioned population and geographical characteristics of these territories. It is important to address inequalities where women often encounter barriers to accessing diagnostic services, lack of financial and physical independence, low literacy and stigma within the home [69,70]. These findings complement what was reported by Robsky et al. [71] and Jenkins et al. [72], who found that a reduction in geographic accessibility is associated with poor outcomes in indicators of diagnosis, adherence and treatment of TB. These areas require a closer look at the event because they can create great variation in cases of TB. Additionally, to achieve the goals established in the End TB strategy and reducing the risk of disease and death, it is very important to expand the screening, diagnosis and treatment of latent mycobacterium tuberculosis infection.

Amazonas, a territorial entity bordering Peru and Brazil, countries considered of high burden at the regional and global level [16], has the highest rates of YYL, YLD and DALYs per TB in the country in men and women. Guainía is one of the five entities with the highest rate of YYL, YLD and DALYs. These two territorial entities have a high rurality and indigenous ethnic population. A possible explanatory route is the social determinants present there, which include poor living conditions, rapid increase in population density [73], food insecurity, ignorance of the disease, low income, poorly ventilated housing, small residences with large numbers of inhabitants [74], low educational level [75] and geographical and administrative barriers [76], all of which intensify the vulnerability to TB. Additionally, their access to health services requires economic resources to take patients to the municipal urban area, which can take several hours given the navigation required through rivers. The high rates of YYL, YLD and DALYs in Risaralda, Atlántico and Meta could have as an explanatory route the co-infection TB-HIV [77], the geographical location that makes them a site of internal and external migration [78], and the fact that it is a reference center for the provision of health services in surrounding regions.

At the subnational level, there are territorial gaps; however, the epidemiological stratification of risk shows us that a large part of the territory is of high load for TB. It is important to mention that policymakers and decision makers must take into account that the burden of disease must be managed by health systems. The burden of TB disease in Colombia comes mainly from premature mortality in men. Given the above, it could be indicated that men do not have full access to monitoring services and remain infectious in the community for considerable periods of time. This is a priority problem that demands special care and necessary improvements of the health system for the attention of this event.

A look at the key risk factors that contribute to the high burden in men, including alcohol and smoking, should generate initiatives aimed at reducing their consumption with stricter public health policies. The window of opportunity for efficient synergies and smart investments in the context of efforts in and for the COVID-19 pandemic could strengthen the global response to TB. The infrastructure for the control of the event, the promotion of better living standards, the alleviation of poverty, the massive campaigns to detect TB and the reduction in comorbidities such as diabetes and HIV infection are priorities. It is important to weigh the role of the first level of care of the health service provider network in such a way as to favor accessibility conditions.

The results of this research can be used to examine the causes of variations at the subnational level and accordingly plan, develop, implement and redirect policies to improve health and eliminate disparities. Colombia does not escape from the universal and geographical influences on health, a fact that reinforces the imperative need to make periodic and regular reports with the methodology of burden of disease, which, in addition to allowing comparability with the rest of the world, supports informed decision making. This helps those responsible for the design and implementation of public policies to explore opportunities for improvement and emulate countries with strategies for information analysis, prevention and control that are working well.

Investment in research and development is needed to identify new and more effective intervention strategies; the necessary changes of health services in terms of technologies, facilities and trained personnel; the development of contextual educational interventions according to cultures, peoples and communities; and to generate treatment guidelines according to the sex of the patient. This will contribute to better health outcomes.

Finally, the research described here is important in filling the knowledge gap that exists in the clear understanding of the burden of TB, which is stipulated in Colombia from simple indicators.

It is suggested, according to the subnational context in this study found, within the framework of primary health care, to strengthen the first level of care. With the above, it contributes to obtaining universal health care, guaranteeing access to health services, especially in dispersed rural and rural areas and prioritizing groups of greater vulnerability such as women and differential populations. For comprehensive care, it is important to create multidisciplinary extra-institutional teams trained in approaching TB and in using technologies that permit timely diagnosis, which allow care to focus on the patient, their family and their community, as well as the diagnosis, treatment and timely follow-up, so that, with this intervention, an improvement in results is achieved, especially to reduce the burden of TB in the country, given the weight of TB mortality in this indicator.

## 5. Limitations

The possible limitations of this research may be related to the use of secondary data, which were solved with the validation process and the control of biases as much as possible. It was not possible to adjust for errors resulting from the underreporting of morbidities.

## Figures and Tables

**Figure 1 tropicalmed-07-00250-f001:**
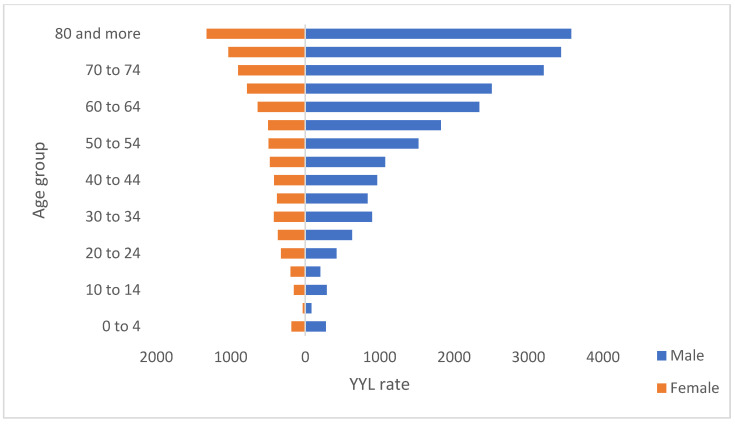
YYL rate by sex an age groups, Colombia, 2010–2018.

**Figure 2 tropicalmed-07-00250-f002:**
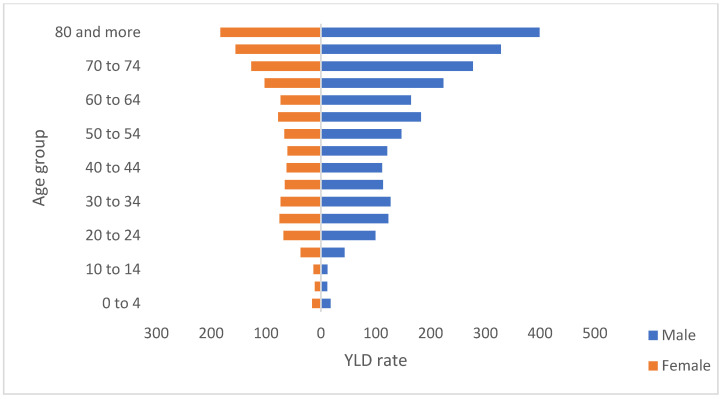
YLD rate by sex an age groups, Colombia, 2010–2018.

**Figure 3 tropicalmed-07-00250-f003:**
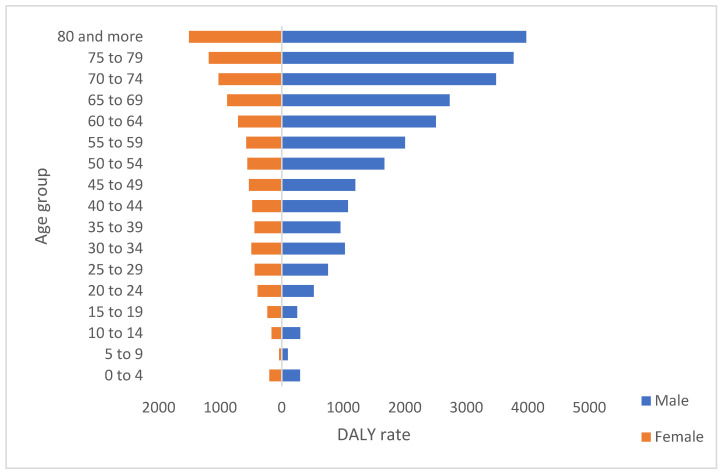
DALY rate by sex an age groups, Colombia, 2010–2018.

**Table 1 tropicalmed-07-00250-t001:** Tuberculosis mortality and mobility by sex and territorial entities, Colombia, 2010–2018.

Territorial Entities	Morbility		Mortality	
	Total Female	Total Male	Total Female	Total Male
AMAZONAS	659	1064	27	45
ANTIOQUIA	21,530	35,208	1191	2152
ARAUCA	791	1304	43	78
ATLANTICO	7727	12,626	665	1165
BOGOTA	8000	13,410	743	1355
BOLIVAR	3119	5079	232	392
BOYACA	893	1528	74	142
CALDAS	2460	4125	184	341
CAQUETA	1271	2095	83	156
CASANARE	961	1585	54	102
CAUCA	2263	3689	132	232
CESAR	2412	3915	182	326
CHOCO	2185	3364	83	138
CORDOBA	2101	3429	184	331
CUNDINAMARCA	2874	5014	188	379
GUAINIA	105	168	3	4
GUAVIARE	280	469	9	18
HUILA	2628	4463	191	356
LA GUAJIRA	2336	3498	183	280
MAGDALENA	2124	3436	160	279
META	3480	5783	256	462
NARIÑO	1400	2254	140	240
NORTE DE SANTANDER	3554	5942	286	528
PUTUMAYO	881	1437	25	42
QUINDIO	1956	3308	129	235
RISARALDA	4139	6699	273	499
SANANDRES	116	195	9	17
SANTANDER	4726	7915	383	731
SUCRE	603	991	41	70
TOLIMA	3541	5943	277	492
VALLE	15,738	25,845	1072	1890
VAUPES	113	190	8	14
VICHADA	186	302	2	5
COLOMBIA	107,152	176,273	7512	13,496

## Data Availability

The data were obtained from SISPRO in Colombia. The anonymized data were validated by the Ministry of Health and Social Protection of Colombia, after consultation requested and made to that institution. The information can be consulted at https://orfeo.minsalud.gov.co/orfeo/consultaWebMinSalud/ (accessed on 20 April 2022), file No. 202142301807932 dated 23 September 2021, time: 03:07:03 and verification code 412cc.

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
