# Peer review of "Tuberculosis Disability Adjusted Life Years, Colombia 2010–2018"

_tropicalmed, 2022, doi:10.3390/tropicalmed7090250_

Round 1

Reviewer 1 Report

The equations described in the statistical analysis section need more precise descriptions and should be numbered when appropriate. The authors use different fonts for the equations and there is an equation in the Spanish language Page 4, line 183. AVD Total or AVAD are not described in the text.

Figures A1.1, A1.2 and A1.3 have text in the Spanish language.

Figure A2 has text in the map that is not readable definitely needs a higher resolution.

Author Response

Dear doctors,

We welcome submission of peer review comments, which we review. We believe that the attached adjusted document reflects the focus of the important issues that needed to be addressed to be acceptable. We request that you reconsider it for publication. Given the above, we comment on each of the observations sent.

Reviewer 2 Report

I read with great interest the paper. I find it well wrote and with good idea research. Furthermore data from Colombia are scarse and it is very interesting read it.

Below my suggestions:

1. Introduction: very well wrote. Only minor English editing is required.

Clarify better the TB indicator of Colombia (Tb pansesible, TB monoresistance, TB death, TB sputum positive, TB HIV confection) 

2. Methods and results: clear and both figure and tables are adequate to content of paper. 

3. Discussion: here I believe that you need improve underlining some issue:

3a. Add information and include more data on co-infection tb- COVID (see Concurrent cavitary pulmonary tuberculosis and COVID-19 pneumonia with in vitro immune cell anergy. Infection. 2021 Oct;49(5):1061-1064. doi: 10.1007/s15010-021-01576-y. ) and also describe the role that COVID pandemic had on TB diagnostic delay with increase of clinical severity and worse outcome (see Increase in Tuberculosis Diagnostic Delay during First Wave of the COVID-19 Pandemic: Data from an Italian Infectious Disease Referral Hospital. Antibiotics (Basel). 2021 Mar 8;10(3):272. doi: 10.3390/antibiotics10030272)

In addiction, line 373-374 is relevant issue, the universal coverage of care and diagnosis. Stress this item.

Moreover, add limitation section

In conclusion, give some global health proposal that came from your study and with strategy can be effectives to improve tb burden

Author Response

(The authors gave the same response as above.)
